# No preconscious attentional bias towards itch in healthy individuals

**Jennifer M. Becker**[1]*, **Henning Holle**[2], **Dimitri M. L. van Ryckeghem**[3,4,5], **Stefaan Van Damme**[3], **Geert Crombez**[3], **Dieuwke S. Veldhuijzen**[1], **Andrea W. M. Evers**[1,6], **Ralph C. A. Rippe**[7], **Antoinette I. M. van Laarhoven**[1]

1 Health, Medical and Neuropsychology Unit, Institute of Psychology, Faculty of Social and Behavioral Sciences, Leiden University, Leiden, The Netherlands, 2 Department of Psychology, Faculty of Health Sciences, University of Hull, Hull, United Kingdom, 3 Department of Experimental- Clinical and Health Psychology, Faculty of Psychology and Educational Sciences, Ghent University, Ghent, Belgium, 4 Section Experimental Health Psychology, Department of Clinical Psychological Science, Faculty of Psychology and Neuroscience, Maastricht University, Maastricht, The Netherlands, 5 Research Unit INSIDE, Institute of Health and Behavior, Faculty of Humanities and Social Sciences, University of Luxembourg, Esch-sur-Alzette, Luxembourg, 6 Medical Delta professor Heatlhy Society, Leiden University, TU Delft, Erasmus Rotterdam, The Netherlands, 7 Research Methods and Statistics, Institute of Education and Child Studies, Faculty of Social and Behavioral Sciences, Leiden University, Leiden, The Netherlands

* j.m.becker@fsw.leidenuniv.nl

**Data Availability Statement:** All relevant data is available in the Dataverse database (https://doi.org/10.34894/PONM1X).

## Abstract

Rapidly attending towards potentially harmful stimuli to prevent possible damage to the body is a critical component of adaptive behavior. Research suggests that individuals display an attentional bias, i.e., preferential allocation of attention, for consciously perceived bodily sensations that signal potential threat, like itch or pain. Evidence is not yet clear whether an attentional bias also exists for stimuli that have been presented for such a short duration that they do not enter the stream of consciousness. This study investigated whether a preconscious attentional bias towards itch-related pictures exists in 127 healthy participants and whether this can be influenced by priming with mild itch-related stimuli compared to control stimuli. Mild itch was induced with von Frey monofilaments and scratching sounds, while control stimuli where of matched modalities but neutral. Attentional bias was measured with a subliminal pictorial dot-probe task. Moreover, we investigated how attentional inhibition of irrelevant information and the ability to switch between different tasks, i.e., cognitive flexibility, contribute to the emergence of an attentional bias. Attentional inhibition was measured with a Flanker paradigm and cognitive flexibility was measured with a cued-switching paradigm. Contrary to our expectations, results showed that participants attention was not biased towards the itch-related pictures, in facts, attention was significantly drawn towards the neutral pictures. In addition, no effect of the itch-related priming was observed. Finally, this effect was not influenced by participants' attentional inhibition and cognitive flexibility. Therefore, we have no evidence for a preconscious attentional bias *towards* itch stimuli. The role of preconscious attentional bias in patients with chronic itch should be investigated in future studies.

**Funding:** This research is supported by a Leiden University Fonds (www.luf.nl) project grant (CWB 7510 / 21-03-2017 / dDM) to A.I.M. van Laarhoven and an Innovative Scheme (Veni) grant (451-15-019) of the Netherlands Organization for Scientific Research (NWO, www.nwo.nl), granted to A.I.M. van Laarhoven. H. Holle was supported by a grant from the Psoriasis Association(www.psoriasis-association.org.uk) (award number ST2/18,). The funders had no role in study design, data collection and analysis, decision to publish or preparation of the manuscript.

**Competing interests:** The authors have declared that no competing interests exist.

## Introduction

Somatosensory stimuli, such as itch or pain, are common experiences in everyday life, signaling potential danger in the environment that may be harmful to the body. These perceptions may lead to behavioral adaptation in attempting to avoid further contact with the source of itch and pain to protect bodily integrity. This is an adaptive process, called attentional bias (AB), that is defined as preferential attention allocation towards threat-related stimuli compared to neutral stimuli [1, 2].

AB can occur at different stages in time of the attentional processing. Posner suggested the existence of an initial *alerting* response, elicited by an external stimulus, i.e., perceiving something in the environment, which then leads to *orienting* of attention towards this stimulus, and lastly *executive control* which determines how we engage with the stimulus [3, 4]. Meta-analyses on AB in the context of pain, which shares many similarities with itch [5–7], confirmed that different presentation times of pain-related information lead to different findings, suggesting that the allocation of attention may differ over time [1, 8]. These analyses showed significant attentional bias towards pain for conscious processing between 500-1000ms, while there is limited evidence for shorter(<500ms) or longer presentation times (>1000ms) [1, 8]. However, this might also be due to a very limited amount of studies, especially in the preconscious processing range [9–12].

Regarding itch-related stimuli, research on how attention fluctuates over time is absent. Research so far has focused on conscious (500ms presentation) processing [8, 13, 14], showing an AB towards itch-related pictures in healthy individuals [13]. This finding was however not replicated in a later study performed in a healthy sample using itch-related pictures and words [14]. As available studies only tapped into late orienting towards- and engaging with itch-related stimuli, it remains unclear whether attention is preconsciously and automatically drawn towards itch. It is important to gain more insight in this early phase of alerting and early orienting as fast and automatic processing is found to be important for the protective function of itch [5–7].

Furthermore, there are indications that people who are dealing with itch on a regular basis, i.e., patients with chronic itch, process itch-stimuli differently [15, 16]—a parallel process was already suggested for pain [1, 8]. However, reacting to itch-stimuli in our environment is evolutionarily useful for everyone alike, i.e. we all want to avoid potential harm. In addition, it seems reasonable that dealing with itch on a daily basis might enhance the stimuli's relevance and saliency [17], which might in turn enhance an AB towards (representations of) itch. This raises the question whether an enhanced relevance and saliency of itch is required before individuals show an AB towards itch.

Overall, there is a mixed pattern of results regarding a conscious AB towards itch. One possible explanation for mixed findings could be the influence of individual characteristics which modulate attention to itch. In addition to self-reported characteristics (e.g., catastrophizing about itch) [1, 13–15], it might be that executive functions can influence an AB towards potentially harmful sensations [3, 18–20]. For instance, attentional inhibition of irrelevant stimuli is a necessary component of AB, e.g., there are more things in our environment than only the itch-related stimulus which compete for attention. Furthermore, after perceiving the itch-related stimulus, switching between different demands (i.e., cognitive flexibility) is necessary to adapt our behaviour: from the external stimulus towards the actual itch-unrelated behaviour. Studies on these characteristics are scarce, with some findings indicating that attentional control (related to cognitive flexibility) is negatively associated with AB towards pain [21–23]. In the context of itch, one study is performed showing no association between AB towards itch and attentional inhibition [14]. However this study did also not find evidence for an association between AB towards pain and attentional inhibition.

The aim of the current study was to investigate the existance of an AB towards subliminally presented itch-related pictures in a healthy sample using a dot-probe task, which measures attention towards an itch-related- compared to a neutral stimulus. Implicit priming was used in half of the sample to investigate possible enhancement of the relevance and saliency of itch in the healthy sample before AB towards itch was measured. We hypothesized that the participants would show an AB towards the itch-related pictures, compared to neutral pictures. Second, we assumed that AB towards itch would be greater after itch-priming compared to control-priming. Lastly, we explored whether individuals' attentional inhibition and cognitive flexibility, as assessed by flanker- and task-switching paradigms, respectively, as well as, several self-reported itch-related cognitions, could predict an AB towards itch.

## Materials and methods

### Participants

Altogether, 128 healthy volunteers were included, an due to a lack of earlier research in this area this was based on an estimated medium effect size (Cohen's d = 0.5) in a between subjects design, an alpha of 0.05, and a power of 0.80. One participant had to be excluded because testing appeared to be done twice with the same person, resulting in a sample of 127. Participants needed to be aged between 18 and 35 years and needed to have normal vision (if applicable, corrected with contact lenses, but not with glasses due to eye-tracking measurements). Participants were excluded if they had any (history of) psychological (e.g., ADHD) or medical (e.g., epilepsy, eczema or rheumatoid arthritis) diagnosis; if participants had dyslexia or were color blind, or if they were regular illicit drug users. Recruitment took place within Leiden University, i.e., posters at the faculty and on the university research participation system (SONA Systems Ltd., Tallinn, Estonia) and via social media. All participants gave written informed consent and data was processed in a pseudonymized manner. The Psychology Research Ethics Committee (Leiden University, the Netherlands) approved the study (CEP18-0514/254).

### Procedure

Information about the procedure was provided digitally and after online registration. Communication went either in Dutch or in English. The experimental lab session took place at the Faculty of Social and Behavioural Sciences, Leiden University and took approximately one hour, see Fig 1 for an overview. Information about all procedures were repeated verbally upon arrival in the lab. However, to warrant the subliminal design of the study, participants were told that the sensitivity of different senses would be assessed without mentioning itch specifically. Furthermore, in- and exclusion criteria were checked, whereafter informed consents were signed and participants filled in a short questionnaire on current depression-, anxiety- and stress-levels. Thereafter, participants were randomly allocated to either an itch-priming or control-priming group, based on random number generation (Microsoft Excel, Redmond, Washington, United States), stratified by gender and handedness. After the itch- or control-priming procedure, participants completed a subliminal dot-probe task to measure preconscious attentional bias towards itch pictures, followed by a stimulus-awareness check task. Afterwards, a Flanker task to measure attentional inhibition and a cued-switching task to measure cognitive flexibility were completed; the order was counterbalanced across the sample. Responses were given with the index fingers of both hands. Subsequently, several self-report questionnaires related to the experience of itch were filled in on a computer. Lastly, participants were debriefed about the aim of the study and received monetary reimbursement (€7.50) or course credits for participation.

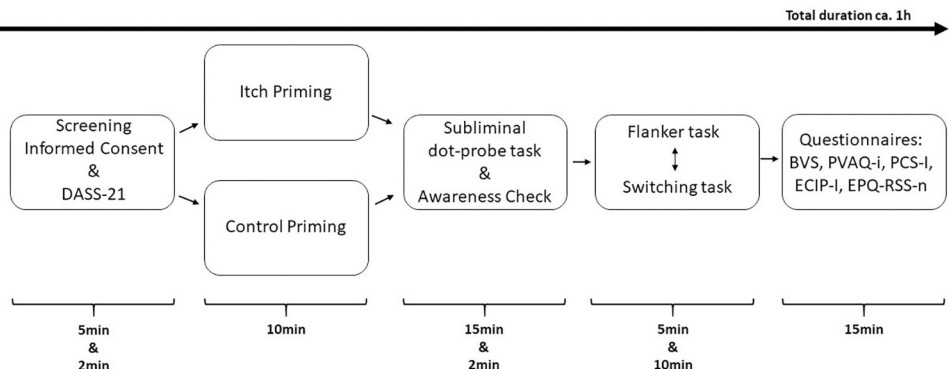

**Fig 1. Study design.** Overview of the procedure during the lab session. *Note*: DASS-21 = Depression, Anxiety, and Stress Scale- short form; BVS = Body Vigilance Scale; PVAQ = Pain Vigilance and Awareness Questionnaire -adjusted for itch; PCS = Pain Catastrophizing Scale -adjusted for itch; ECIP = Experience of Cognitive Intrusions of Pain Scale -adjusted for itch; EPQ-RSS = Neuroticism Scale of Eysenck Personality Questionnaire–revised short form.

## Technical set-up

Tasks were presented on an Iiyama HM703UT A Vision Master Pro 413 CRT monitor (17 inch) with a refresh rate of 100Hz and a resolution of 1024x768px. All attention tasks, i.e., dot-probe task, Flanker task, switching task and the awareness check, were administered in E-Prime 2.0 (Psychology Software Tools, Inc., Sharpsburg, USA) and responses were collected with custom-made response buttons on the right and left side of the table, attached to a Serial Response Box (Psychology Software Tools, Inc., Sharpsburg, USA). Questionnaires were presented with Qualtrics (Provo, Utah, USA) on the computer. Eye-movements were measured during the Dot-Probe task by means of a Tobii Pro X3-120 Eye Tracker (Tobii AB, Danderyd, Sweden). The eye-tracker was attached to the table in front of the screen. Participants were asked to put their head in a chin rest in front of the computer screen during all tasks to guarantee a constant distance towards the screen and the eye-tracker (78cm and 71cm, respectively).

## Priming

The sample was split into an itch-priming and a control-priming group. The priming consisted of three mechanical and three auditory stimuli for both, the itch group and the control group (see specifications below). The stimuli for the itch-group were selected to induce itchiness and therefore may trigger attention to itch, whereas the control stimuli were assumed not to induce any itchiness or attention to itch.

Participants described the experience of each stimulus on six adjectives adapted from the McGill Pain Questionnaire (Part 2) [24]. These descriptors were *itchy* as the variable of main interest and five other descriptors, namely *bothersome*, *painful*, *light*, *pleasant*, and *unpleasant*. All adjectives appeared in random order after each stimulus, embedded in the question "*Please rate how [adjective] you perceived the stimulus on a scale from 0 (does not describe my experience at all) to 4 (describes exactly my experience)*".

**Mechanical stimuli.** During the itch-priming, three Touch Test Evaluators (filaments of 4.08, 4.17 and 4.31 mN, consistently in this order; Stoelting, North Coast Medical, Gilroy, California, USA) were pressed on the skin three consecutive times for one second each [25]. During the control-priming, one steady stroke for about 1s was applied with a Somedic brush (MRS, Heidelberg, Germany) over a length of 1-2cm on the forearm of the participant [26]. This was repeated three times to match the Touch Test Evaluators procedure in the itch-priming group.

**Auditory stimuli.** Three scratching sounds [27], with a duration of 20s, were presented as itch-priming. The itch sounds comprised the sound of scratching different body parts (i.e., arm, arm pit and beard), used in an earlier study on itch contagiousness [27]. Three itch-unrelated control sounds matched in pitch and length were presented as control-priming. The control sounds were recorded for the purpose of this study and consisted of everyday life noises that would not be recognized too easily (rolling a plastic ball over a table, squishing a plastic bag and rummage about a box of foam stickers). Face validity of these new control stimuli was assessed by the research team. Overall loudness of each audio clip was normalized using PRAAT (https://www.fon.hum.uva.nl/praat/). The auditory stimuli were presented in random order within one group (itch vs. control).

## Subliminal dot-probe task

A subliminal Dot-probe paradigm with a validated set of 40 picture pairs was used [14], with 40 itch-related pictures showing someone scratching (itch-pictures). Half were paired with neutral pictures depicting human skin without scratching gestures (skin-pictures) and half being paired with neutral objects (object-pictures). The pairs stayed constant across the task. In total, the task consisted of 320 trials, preceded by 24 practice trials with only skin-picture— object-picture pairs not used during the actual task.

The trial sequence consisted of three displays, as can be seen in Fig 2. First, two pictures appeared, one in the lower and one in the upper part of the screen, followed by two masks at the same location as the original pictures to further inhibit conscious processing of the pictures, i.e., backward masking was employed [28], and lastly a target. The target consisted of two dots to which a response by button press was needed. Orientation of the dots (horizontally vs. vertically) and button side (left vs. right) was counterbalanced across participants, i.e., press right for horizontal dots and left for vertical dots or vice versa. If the dots appeared at the same location as the itch-picture this constituted a congruent trial, whereas if the dots appeared at the same location as the neutral-picture this constituted an incongruent trial. Across the whole task, the itch-pictures appeared in both locations, with both dot orientations and as an incongruent and as a congruent trial. Breaks of 20s were inserted after every 40 trials. Reaction times and accuracy to respond to the orientation of the targets were measured. This task took approximately 15 minutes to complete.

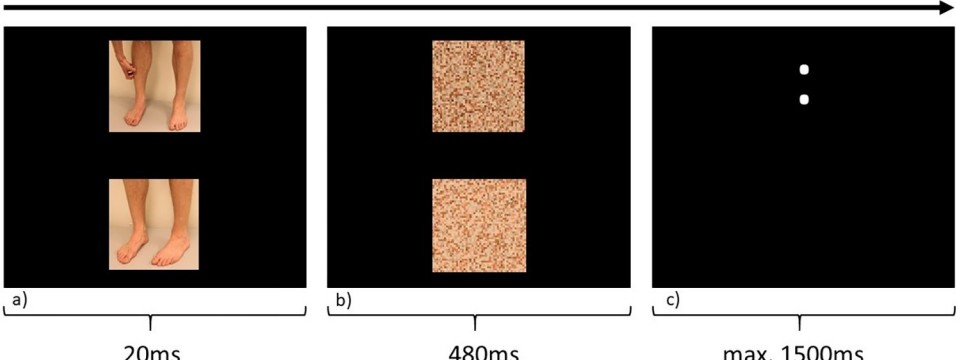

**Fig 2. Dot-probe task.** One trial of the subliminal Dot-Probe task showing a trial with an itch-picture and a skin-picture as control (a) with their corresponding masks (b). The target (c) is presented in the same location as the itch-picture (until button press), making this trial a congruent trial. Additionally, in the middle of the screen, a fixation cross is shown in-between trials (500ms).

**Stimuli and display configurations.** The pictures and the masks were 192x192px which was 25% of the height of the screen (1024x768px screen resolution). The masks were made by dividing the pictures into 4x4px cubes and shuffling them randomly into a new 192x192px picture (MATLAB Release 2017b, The MathWorks, Inc., Natick, Massachusetts, United States). In this way, each picture had its own mask, identical in color and brightness. The pictures and masks appeared at the 20% and 80% height position on the screen and the fixation point at 50% height; all stimuli were in the middle of the screen (50% width). Pictures were presented for 20ms, masks for 480ms and the target dots with a maximum response window of 1500ms (see Fig 2).

## Awareness check

As an objective awareness check, a forced-choice paradigm was employed [28]. On each trial, participants were presented with one new picture and one picture that was subliminally shown during the dot-probe task. Twenty-five percent of the itch-pictures (10), skin-pictures (5) and object-pictures (5) that were shown during the dot-probe task were used for all participants, resulting in 20 trials. The new pictures came from the same validated set and pairs were matched in color and brightness [14]. Participants indicated which one of the two pictures they thought they had seen earlier. If the previously shown pictures were selected at chance level, it was assumed that the pictures were not consciously processed during the dot-probe task. There was no time limit and only accuracy to select the previously shown pictures was measured. Therefore, an accuracy level around 0.5 would indicate that the previously shown pictures were detected at chance level. In addition, as a subjective awareness check, participants were asked orally if they noticed something special during the dot-probe task and this was tracked with 'yes' or 'no' as the outcome. If this was answered with 'yes', participants were asked what they noticed and if they mentioned pictures (Did you see any pictures?), this was recorded as 'yes', as well ('no', if answer did not contain pictures). The total awareness check took approximately two minutes to complete.

## Attentional switching task

A cued attentional switching task was used to assess cognitive flexibility [29]. In this task, participants followed two different instructions. On each trial, first a cue appeared to indicate which of two instructions needed to be followed during this trial. One instruction indicated that participants had to identify if a target number shown in the middle of the screen (i.e., 1, 2, 3, 4, 6, 7, 8, 9) was above (right button) or below (left button) five and the other instruction indicated that they had to identify if the number was odd (left button) or even (right button). Two different trial types can be distinguished, depending on whether the trial before had the same instruction (repeat-trials) or the other instruction than the trial before (change-trial). In total, the task consisted of 200 trials (100 trials per instruction type, randomly presented) plus 16 practice trials, with a break halfway. Reaction times and accuracy to respond to the target number were measured. The task took approximately 10 minutes to complete.

## Flanker task

A Flanker task was used to measure attentional inhibition [29, 30]. During this task, a string of five numbers appeared on the screen, consisting of '4' and '2', with the middle number being the target. Participants were asked to identify the target as being a two (left button) or a four (right button). The flanking numbers on both sides of the target were either the same as the target (i.e., congruent) or not (i.e., incongruent). The task consisted of 120 trials (50% congruent, 50% incongruent) plus 8 practice trials at the beginning, with a break halfway through the

task. Reaction times and accuracy to respond to targets were measured. The task took approximately 5 minutes to complete.

## Eye-tracking measurements

Eye-movements were measured during the dot-probe task only. During the task, it was measured whether the participant's eyes were positioned at the Area-of-Interest (AoI) with a sampling rate of 120Hz. AoI's were specified as the area of both pictures that were shown during the task (see *Stimuli and display configurations*). Data was pre-processed with the PhysioDataToolbox AiO Hit Analyzer [31]. Data was extracted for each trial of the task, with one variable for hit count on the itch-picture and one variable for hit count on the neutral-picture. A hit is counted whenever the participant looked at the AoI.

## Self-report questionnaires

Next to questions about current itch, pain, and fatigue (Numeric rating scales from 0 (not at all) to 10 (worst imaginable)), the following concepts were measured: attention to bodily sensations by the Body Vigilance Scale (BVS) [32] adapted to the current purpose by replacing two items on derealization with itch and pain which were used individually during analyzes; neuroticism by the EPQ-RSS (Eysenck Personality Questionnaire- revised short form): Subscale neuroticism (and subscale extraversion as filler items) [33]; Vigilance towards itch by the Pain Vigilance and Awareness Questionnaire [34] adjusted for itch (PVAQ-I) [14]; catastrophizing about itch by the Pain Catastrophizing Scale [35], adjusted for itch (PCS-I) [36]; and how much itch intrudes one's thoughts by the Experience of Cognitive Intrusion of Pain scale [37], adjusted for itch (ECIP-I) [14]. Lastly, depression, anxiety and stress were measured with the short version of the Depression, Anxiety and Stress Scale (DASS-21). Completion of all questionnaires took approximately 15min, the DASS-21 (2min) before testing and the remaining questionnaires thereafter (13min; see Fig 1).

## Statistical analyses

E-prime data were extracted with E-Prime E-DataAid 3.0 (Psychology Software Tools, Inc., Sharpsburg, USA). For the Dot-Probe task, reaction times (RT, in ms), accuracy, congruency (congruent vs. incongruent), group (priming vs. control), trial number (1 to 320) and trial type (neutral skin vs. neutral object picture) for all individual experimental trials (i.e., without practice trials) were extracted. For the Flanker task, mean RT for congruent and incongruent trials were extracted separately, only including correct trials of the experimental trials with RT > 150ms. In the same way, mean RT for change- and repeat-trials of the Attentional Switching task were extracted. A Flanker index was calculated to use as a predictor describing general attentional inhibition ($RT_{incongruent} - RT_{congruent}$), with positive values indicating greater ability to suppress goal-unrelated responses [20]. Switch costs were calculated to use as a predictor interpreted as general cognitive flexibility ($RT_{change} - RT_{repeat}$), where positive values indicate a greater ability to shift attention from one task to another [38].

Furthermore, mean accuracy was extracted from E-Prime for the objective Awareness check, as well as the individual answers to the subjective awareness questions (i.e., yes or no). Concerning the ratings of the priming material (itch and control), mean ratings for all six adjectives per category (mechanical and auditory) were extracted. Questionnaire data was extracted from Qualtrics and total scores and reliability for the different questionnaires were calculated with SPSS version 23 (IBM Statistics for Windows, Armonk, NY, USA). Missing items of the PVAQ-I (10 participants, one item each) were imputed using the mean of all other items of the corresponding participant. Subsequent statistical analyses were done with R

version 4.0.3 [39]. All tests were done with $\alpha \leq 0.05$ and descriptive results were given as mean and standard deviation, if not indicated otherwise.

Reliability of the Dot-Probe task was assessed with the R package 'splithalfr' [40]. First, mean RT for congruent and incongruent trials were calculated for every participant. Second, Monte Carlo splitting was used to get 5000 split-half's of the sample of mean congruent RT and mean incongruent RT, separately. Lastly, these samples were used to estimate Spearman-Brown coefficients as an estimate of reliability. The mean coefficient and the range of all coefficients were reported. Reliability of an AB index (mean incongruent RT–mean congruent RT) was calculated in the same way. For the reliability analyses, only participants with an accuracy level above 0.70 were included [13, 14].

**Manipulation checks.** Concerning the priming manipulation, a one-way Analysis of Variance (ANOVA) was done for each outcome rating (i.e., itchy, painful, bothersome, light, unpleasant, pleasant), separately for the mechanical and the auditory stimuli, to check for group differences (priming vs. control group). Due to violations of normality, the ANOVA was done with bootstrap (1000 samples) of the residuals (R package "lmboot") [41]. For the objective awareness measure, single proportion tests were used to assess whether picture selection deviated from chance level (50%) by the overall sample. For the subjective awareness questions, frequencies for answering 'yes' and 'no' to the questions were calculated. Lastly, paired sample t-tests were employed to assess whether congruent and incongruent trials differed significantly during the Flanker task and whether change and repeat trials differed significantly during the Attentional Switching task, both for the sample as a whole.

**Preprocessing.** In line with previous work on AB using a dot-probe task, trials with R <150ms were excluded from the main analyses [13, 14]. Additionally, one participant had to be excluded due to an excessive amount of missing data due to technical issues during data collection. Lastly, inspection of the raw data showed one outlier on the Flanker index and to eliminate any possible bias within this participant's responses, this participant was excluded from the main analyses of AB. Altogether 0.02% of the data was excluded from the main analyses.

Inspection of the raw eye -tracking data showed that most of the eye-tracking data points were zero (69,7%) which means that during these trials participants did not fixate on one of the two pictures at all. Likewise, total fixation duration was zero or very close to zero during most trials. Due to this, we decided to omit analyses of the eye-tracking data.

**Multilevel model of AB.** Due to the repeated measures design of the Dot-Probe task with trials (level 1) nested within subjects (level 2), multilevel models were estimated with the mixed models R packages 'lme4' and 'lmerTest' [42, 43]. RT data typically do not follow a normal distribution and therefore initially a generalized linear model was used with an inverse Gaussian link function [44]. Inspection of its results, a visual check of the empirical RT distribution in the current sample, as well as results from a linear multilevel model using a normal distribution showed that results of the linear multilevel model did not substantially differ from the results using an inverse Gaussian link function. As the linear model with a non-inverse Gaussian link function is simpler to compute and to interpret (e.g., estimates on original scale), we decided to use a linear multilevel model with a Gaussian link function for the analyses.

Our hypotheses were tested with a multilevel model with Dot-Probe task RT data as outcome and random intercepts for subject to account for the repeated measures design and random intercepts for trial number to account for an expected learning curve during the task (i.e., participants are getting better at the task over time). Models were built according to the hypothesis of the study and in case of convergence issues, choices were based on the priority of the research questions. As of main interest, the fixed effect of accuracy, congruency and priming group were added to the model (Model 1). Accuracy was included to control for its effect

**Table 1. *M (SD)* for the ratings of the mechanical and auditory stimuli for the priming group (*n* = 63) and the control group (*n* = 64).** *P*-values with bootstrapped residuals are reported to indicate significant group differences due to skewd distributions. Parametric effect sizes ($\eta^2$) are reported. 'Itchy' as the descriptor of main interest is printed bold.

| | Mechanical priming stimuli | | | | Auditory priming stimuli | | | |
|---|---|---|---|---|---|---|---|---|
| | Priming | Control | *p* | $\eta^2$ | Priming | Control | *p* | $\eta^2$ |
| **Itchy** | **1.13 (1.13)** | **0.71 (0.91)** | **0.025** | **0.04** | **1.21 (1.21)** | **0.77 (0.92)** | **0.018** | **0.04** |
| Painful | 0.35 (0.52) | 0.24 (0.51) | 0.246 | 0.01 | 0.55 (0.79) | 0.32 (0.56) | 0.071 | 0.03 |
| Light | 2.58 (1.11) | 3.02 (0.90) | 0.020 | 0.05 | 1.13 (0.92) | 1.40 (0.92) | 0.101 | 0.02 |
| Bothersome | 0.81 (0.84) | 0.50 (0.77) | 0.033 | 0.04 | 1.98 (1.04) | 1.66 (0.94 | 0.075 | 0.02 |
| Pleasant | 1.49 (1.30) | 2.21 (1.25) | 0.003 | 0.08 | 0.75 (0.78) | 1.11 (0.88) | 0.019 | 0.05 |
| Unpleasant | 0.86 (0.89) | 0.51 (0.80) | 0.029 | 0.04 | 1.98 (1.05) | 1.45 (1.02) | 0.006 | 0.06 |

on RT, i.e., it might be assumed that participants did not attend well to the task at all whenever they gave a wrong response. The hypothesis of whether the participants display an AB towards itch was tested with the effect of congruency. The hypotheses that AB would be greater after itch priming compared to control priming was investigated with the congruency by group interaction effect. In a next step, the Flanker index and switch cost were added as predictors (fixed effects) to the model to investigate their effect on an AB towards itch (Model 2) and their interaction with congruency was explored to investigate their specific effect on AB (Model 2a and 2b). Lastly, participants' scores on the self-report questionnaires (i.e., body vigilance, neuroticism, itch vigilance and awareness, itch catastrophizing and cognitive intrusions by itch) were added as covariates to the model to control for their possible effect on the outcome and get more precise estimates of the effects of interest (Model 3). QQ-plots of the residuals of the final model were inspected for possible bias in the estimation.

## Results

### Participants

The sample consisted of 127 participants, 107 females and 20 males with a mean (*M*) age of 21.9 years (standard deviation (*SD)* = 2.5). Participants were mostly right-handed (113 right vs. 14 left). Descriptive statistics for the self-report questionnaires can be found in S1 Table and correlations with the AB-index can be found in S2 Table. The priming group and the control group did not differ significantly on any of the demographic and self-report variables (e.g., age, gender, Flanker Index, itch vigilance), all *p* > 0.05.

### Manipulation checks

**Priming.** Descriptive statistics and test results of the priming manipulatiocan be found in Table 1. Concerning the descriptor of main interest–'itchy', the priming group rated both, the mechanical and the auditory stimuli as significantly more itchy than the control group, see Table 1.

**Awareness.** Overall, the whole sample selected the picture that was shown during the subliminal Dot-Probe task compared to a new, unused picture, with a mean accuracy of 0.49 (*SD* = 0.13) and the single proportion test showed that this did not significantly differ from 50%, *p* = 0.592. Furthermore, for the subjective awareness questions, 50 participants indicated that they noticed something during the Dot-Probe task and 45 of these participants also indicated that they saw some other pictures besides the scrambled masks that were used during the task, but none reported anything related to itch or scratching.

**General attention tasks.** During the Flanker task, participants were, conform expectations, significantly faster during congruent trials ($M$ = 393.39ms, $SD$ = 71.44ms) than incongruent trials ($M$ = 439.14ms, $SD$ = 81.81ms), indicating significant interference by incongruent flanking numbers that needed to be inhibited, $t$ = -4.73, $p$ > 0.001, mean difference ($MD$) = 45.62. During the Attentional Switching task, participants were significantly faster during repeat trials ($M$ = 643.51ms, $SD$ = 187.03ms) than change trials ($M$ = 729.48ms, $SD$ = 234.30ms), indicating that there was a significant switch cost due to switching between sub-task instructions, $t$ = -3.23, $p$ = 0.0014, $MD$ = 85.97.

## Analyses of AB

Reliability of the congruent and incongruent trials of the dot-probe task was high, with a mean Spearman-Brown coefficient of 0.98 (IQR = 0.96; 0.98) for congruent trials and 0.98 (IQR = 0.97; 0.99) for incongruent trials. The mean Spearman-Brown coefficient for the AB index was 0.51 (IQR = 0.45; 0.59). Descriptive statistics for the RT data can be found in Table 2. Model fit can be inspected in S1 Fig.

Model 1 (Table 3) of the multilevel analyses shows that that there was a significant effect of congruency on the outcome RT, indicating that congruent trials are 3.23ms slower than incongruent trials, $t$ (39528.57) = -1.998, $p$ = 0.046. Thus, participants were slower to make orientation judgments to targets appearing in a location previously occupied by an itch picture, as compared to a neutral picture, showing a preconscious tendency to avoid itch pictures. Furthermore, there was no significant main effect or interactions involving the factor group, indicating that priming does not change the difference in reaction times between congruent and incongruent trials.

Model 2 (Table 3) confirmed the significant effect for congruency found in Model 1, $t$ (3953) = -1.998, $p$ = 0.046, controlling for Flanker Index and Switch Cost. Both variables were not significantly related to the outcome. However, when Flanker Index and Switch Cost were both added as an interaction term with congruency (Model 2a, see S3 Table), the significant main effect of congruency disappeared $t$(3953) = -1.076, $p$ = 0.282, while also the Flanker Index by congruency interaction, $t$(3953) = -0.128, $p$ = 0.898, and the Switch Cost by congruency interaction remained not significant, $t$(3953) = 0.098, $p$ = 0.922. Although, the original hypothesis was to include the Flanker Index and Switch Cost as predictors, including them as covariates was explored to further investigate the abovementioned findings. When the Flanker Index by congruency interaction is removed and it is only controlled for the main effect of Flanker index (i.e., it is included as a covariate only) (Model 2b, see S3 Table), a trend towards a significant main effect of congruency returns, $t$(3952) = -1.702, $p$ = 0.089. This means that the non-significant Flanker Index by congruency interaction is collinear to the main effect of congruency, showing that the interaction does not add any new information to the model in addition to the main effect of congruency.

**Table 2. Mean (*SD*) for the reaction time data (ms) of the subliminal dot-probe task for itch per group (priming vs. control) and congruency (congruent vs. incongruent) (*n* = 127).**

|  | Priming Group | Control Group |
|---|---|---|
| Congruent | 468.05 (129.27) | 480.22 (137.93) |
| Incongruent | 466.46 (128.07) | 476.88 (134.58) |
| *AB index* | *1.77 (13.83)* | *2.99 (13.62)* |

Note. AB index = reaction times$_{incongruent}$−reaction times$_{congruent}$

**Table 3. Multilevel analyses with RT as outcome variable for the subliminal dot-probe task for itch: estimates (*ES*) with standard errors (*SE*), significance level (*p-value*) and 95% confidence intervals (95% *CI*) (*n* = 125).**

|  |  | ES | SE | p-value | 95% CI |
|---|---|---|---|---|---|
| Model 1 | (Intercept) | 462.53 | 8.83 | < 0.001 | [445.24, 179.83] |
|  | Accuracy | 19.76 | 2.24 | < 0.001 | [15.37, 24.15] |
|  | Congruency | -3.23 | 1.62 | 0.046 | [-6.40, -0.06] |
|  | Group | -11.88 | 11.99 | 0.324 | [-35.38, 11.62] |
|  | Congruency * Group | 1.52 | 2.28 | 0.505 | [-2.95, 5.98] |
| Model 2 | (Intercept) | 471.40 | 15.14 | < 0.001 | [442.00, 500.87] |
|  | Accuracy | 19.75 | 2.24 | < 0.001 | [15.36, 24.14] |
|  | Congruency | -3.23 | 1.62 | 0.046 | [-6.40, -0.06] |
|  | Group | -11.75 | 11.95 | 0.328 | [-34.98, 11.49] |
|  | Congruency * Group | 1.52 | 2.28 | 0.505 | [-2.95, 5.98] |
|  | Flanker Index | -0.33 | 0.24 | 0.176 | [-0.80, 0.14] |
|  | Switch Cost | 0.07 | 0.07 | 0.327 | [-0.07, 0.21] |
| Model 3 | (Intercept) | 473.90 | 24.38 | < 0.001 | [428.28, 519.50] |
|  | Accuracy | 19.80 | 2.24 | < 0.001 | [15.42, 24.20] |
|  | Congruency | -3.23 | 1.62 | 0.046 | [-6.40, -0.06] |
|  | Group | -11.30 | 12.16 | 0.355 | [-34.04, 11.46] |
|  | Congruency * Group | 1.52 | 2.28 | 0.505 | [-2.95, 5.98] |
|  | Flanker Index | -0.35 | 0.25 | 0.163 | [-0.82, 0.12] |
|  | Switch Cost | 0.04 | 0.07 | 0.570 | [-0.10, 0.18] |
|  | Disengagement Itch | -1.64 | 3.75 | 0.663 | [-8.65, 5.32] |
|  | Disengagement Pain | -0.57 | 3.10 | 0.854 | [-5.23, 6.37] |
|  | Body Vigilance total | -3.66 | 4.21 | 0.386 | [-11.52, 4.20] |
|  | Body Vigilance Itch | 2.64 | 4.16 | 0.527 | [-5.13, 10.41] |
|  | Body Vigilance Pain | 3.38 | 3.20 | 0.293 | [-2.60, 9.36] |
|  | Itch Vigilance & Awareness | -0.61 | 0.62 | 0.326 | [-1.77, 0.55] |
|  | Itch Catastrophizing | -0.14 | 1.24 | 0.907 | [-2.17, 2.46] |
|  | Cognitive Intrusions by Itch | 0.52 | 1.07 | 0.629 | [-2.52, 1.48] |
|  | Neuroticism | 3.44 | 2.12 | 0.108 | [-0.52, 7.40] |

Note. Model fit statistics; Model 1: AIC = 491202.2; Model 2: AIC = 491207.8; Model 3: AIC = 491190.1

Lastly, Model 3 (Table 3) which included several self-report characteristics as covariates, again showed a significant association of congruency with the outcome, *t*(3953) = -1.998, *p* = 0.046.

## Discussion

Because it is assumed that potentially threatening stimuli, including itch, draw attention, the current study investigated whether attentional bias (AB) towards visual itch-stimuli already shows up when stimuli are subliminally presented. In contrast to the hypothesis, healthy participants avoided the preconsciously presented itch-pictures and this effect was not influenced by priming participants with a mild itch stimulus and scratching sounds. Moreover, there was no significant association between preconscious AB towards itch and attentional inhibition or cognitive flexibility. But preliminary findings showed that attentional inhibition might be related to the emergence of an AB. Altogether, this study did not support preconscious orienting of attention towards itch-related stimuli, but rather suggests that healthy individuals orient away from these stimuli.

The finding that peoples' attention is not preconsciously biased towards itch-related pictures, but is actually oriented away, is in contrast with our hypothesis. However, very fast and automatic avoidance of itch-related stimuli can still be explained by its protective function, because the ultimate goal is to avoid the source of the potential threat [5–7]. In addition, scratching is stigmatized [45–48] which could explain why people avoid the itch-related stimuli. Actually, this could be adaptive as long as someone is not directly in contact with someone who is scratching: infection through a picture is not possible. Consequently, orienting away from someone who is scratching is adaptive to avoid direct contact. In addition, seeing someone else scratching could induce disgust in the viewer which would also support avoidance of these stimuli. However, to our knowledge, it has not been studied yet how disgust specifically relates to attention to itch and related stimuli. Yet, research has shown that skin-related disgust plays a role for patients with chronic skin diseases, so a relationship between attention to itch and disgust seems plausible [49]

Nevertheless, the cumulative evidence for a preconscious AB towards- or avoidance of threat related stimuli like itch or pain is limited. It has to be taken into account, that the handful of studies on subliminal processing of pain-related stimuli used different stimuli and some also different paradigms which makes drawing conclusions difficult [9–12]. Beyond the fact that these studies had contradictory findings, two studies measured attentional interference (using a Stroop task) instead of attention towards a location, i.e., orienting towards a stimulus [9, 10] which is a different aspect of attention, although related to AB [3, 50]. Lastly, none of these earlier studies used pictorial stimuli, but used word-stimuli, and research on AB towards pain and threat has shown that results might differ for picture- and word stimuli [1, 8, 51]. Therefore, it is unclear how this might influence preconscious attention. Moreover, the saliency of the aversive content probably differs between itch- or pain-related content, as well as for pictorial representations compared to semantic processing, i.e., words.

In contrast to our expectations, priming with a mild itchy stimulus and scratching sounds did not seem to to enhance relevance and herewith attention to itch in the current study. This was in spite of the fact that the itch-priming stimuli were, as intended, rated as significantly more itchy than the control-stimuli. Possible explanations why the itch-priming did not result in more attention to itch might be that the stimulus was not itchy enough and that more pronounced itch stimuli, like cowhage [52], are needed to heighten the relevance and saliency sufficiently in healthy participants. Hence, participants presumably were not consciously focused on itch which may have hampered the priming's effectiveness to change AB towards itch. Not mentioning the context if itch to participants in the present study is in contrast to most studies that used audiovisual stimuli for the investigation of contagious itch, which as far as we know, explicitly mentioned the relation to itchiness to the participants [27, 53]. Furthermore, the actual task that needed to be executed (identification of dot orientation) was not related to itch and therefore not directly related to the priming, which could also decrease its effectiveness. Lastly, in the current study, the priming and the actual task were in different modalities (feeling and hearing during priming and visual processing during the task) [54].

Investigating cognitive flexibility in relation to AB, which has not been done before, did not yield any significant associations. In line with this, there was also no significant association between AB and attentional inhibition, although exploratory analyses showed, that AB and inhibition share a common factor as their effects overlapped in the current models. This could mean that someone who can generally better inhibit/ignore irrelevant information, is also better in ignoring a task-irrelevant itch-related cue in the environment. This would support the avoidant pattern observed in the current study and underline the fact that itch-related pictures are no direct danger of infection and can be ignored (see above). Nevertheless, as these finding is exploratory and preliminary, future research is needed. There is one other study so far that

investigated an association between AB towards itch and attentional inhibition, but this study did not find a significant overall AB (nor avoidance), which makes interpretation difficult [14]. Nevertheless, attention is a complex phenomenon that is interrelated with other executive functions [18, 20, 55], and accordingly it is recommended not to study AB in isolation. There are indications that attentional control, as a part of executive functions, is compromised in patient groups [56], although not always confirmed [57, 58] which could suggest that executive functions play a bigger role in AB in patient groups than in healthy controls. Regarding the exploratory investigation of possible associations between itch-related cognition and AB towards itch, it is interesting to note that specifically awareness of bodily sensations is negatively related to a lower AB towards itch (see S2 Table). As a low (i.e. negative) index of AB actually indicates avoidance, this could mean that individuals who are more aware of bodily sensations might also be more avoidant of potential bodily harm. But this remains speculation at this point.

The current study has several limitations. First, the sample was very homogeneous with mostly female university students. Second, in the current study the between-group difference in induced itch by using only a mild itch-stimulus was of limited size. Third, due to the subliminal design, there was no baseline itch measurement and also no prove that participants were completely itch-free before participation (e.g., no mosquito bites). Forth, the current subliminal design appeared to be inappropriate to make good use of the eye-tracking data. Future research could aspire to circumvent these limitations by using a more pronounced itch stimulus and additionally also mention the fact that the stimulus is expected to induce itch. In this way, participants would be also consciously primed for itch which could enhance its effects on attentional processing. However, this would be difficult to combine with a preconscious design with deception as used in the present study. A possible solution might be to use longer presentation times, which make the itch-related content visible to the participant and to combine this with a very time-sensitive measurement like eye-tracking and/or electroencephalograms to differentiate between the different stages of attentional processing, e.g., early orienting towards the stimulus compared to late disengagement.

All in all, the current study found preconscious avoidance of itch-related visual stimuli in healthy individuals. Such avoidance might be different from attentional processing of actual somatosensory itch stimuli, but as somatosensory itch is difficult to study preconsciously, our study is a good approximation for a preconscious study. Furthermore, patients with chronic itch need to be investigated in the future because it can be assumed that patient's attention towards itch-related stimuli differs compared to healthy controls which has already been shown for contagious itch which involved attentional processing [59].

## Supporting information

**S1 Table. Self-report questionnaires of individual characteristics (*n* = 127).**
(DOCX)

**S2 Table. Spearman rho (*ρ*) correlations between individual characteristics and the Attentional Bias (AB) index for itch (*n* = 127).**
(DOCX)

**S3 Table. Exploratory analyses of the effect of the Flanker index and switch cost on attentional bias during the subliminal dot-probe task for itch: Estimates (*ES*) with standard errors (*SE*), significance level (*p-value*) and 95% confidence intervals (95% *CI*) (*n* = 125).**
(DOCX)

**S1 Fig. QQ-plot of the residuals of Model 1 for the reaction times outcome of the subliminal dot-probe task for itch.** Inspection of the residual distribution of this QQ-plot to assess model fit and potential bias shows that the fitted models were more accurate for lower values, while being slightly biased upwards for higher values. However, inspecting the QQ-plot of Model 3 (not shown here, but similar to the QQ-plot of Model 1) with more sources of information, even after adding scores of the awareness check and control neutral picture type (skin vs. object), could not reduce the observed small bias for higher values. *Note.* Because all QQ-plots for all the different models are similar, this one is shown as an example for all described models in this study.

(DOCX)

## Acknowledgments

We would like to thank Fabiënne van den Ende, Rose van Oostveen, Annie Nguyen, Klara Bokelmann and Diana Czepiel for their support with data collection and Rosanne van den Berg for recording the control priming sounds with us.

## Author Contributions

**Conceptualization:** Jennifer M. Becker, Henning Holle, Dimitri M. L. van Ryckeghem, Stefaan Van Damme, Geert Crombez, Andrea W. M. Evers, Antoinette I. M. van Laarhoven.

**Data curation:** Jennifer M. Becker.

**Formal analysis:** Jennifer M. Becker, Ralph C. A. Rippe.

**Funding acquisition:** Antoinette I. M. van Laarhoven.

**Methodology:** Jennifer M. Becker, Henning Holle, Dimitri M. L. van Ryckeghem, Stefaan Van Damme, Geert Crombez, Ralph C. A. Rippe, Antoinette I. M. van Laarhoven.

**Project administration:** Jennifer M. Becker, Antoinette I. M. van Laarhoven.

**Supervision:** Dieuwke S. Veldhuijzen, Antoinette I. M. van Laarhoven.

**Writing – original draft:** Jennifer M. Becker.

**Writing – review & editing:** Henning Holle, Dimitri M. L. van Ryckeghem, Stefaan Van Damme, Geert Crombez, Dieuwke S. Veldhuijzen, Andrea W. M. Evers, Ralph C. A. Rippe, Antoinette I. M. van Laarhoven.

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
