## [Decision Letter · Decision Letter 0]

2 Jun 2022

PONE-D-22-05721No preconscious attentional bias towards itch in healthy individualsPLOS ONE

Dear Dr. Becker,

Thank you for submitting your manuscript to PLOS ONE. After careful consideration, we feel that it has merit but does not fully meet PLOS ONE’s publication criteria as it currently stands. Therefore, we invite you to submit a revised version of the manuscript that addresses the points raised during the review process. Editor comment. I found one expert reviewer to comment on your work. I also read the manuscript myself. Detailed suggestions can be found at the bottom of this letter. The referee considers your work important overall and suggested some points to consider before final publication. I would invite you preparing a revision of your work that addresses all concerns together with a cover letter that provides point-by-point replies.

We look forward to receiving your revised manuscript.

Kind regards,

Michael B. Steinborn, PhD

Section Editor

PLOS ONE

Journal Requirements:

This research is supported by a Leiden University Fonds project grant (CWB 7510 / 21‐03‐2O17 / dDM ) to A.I.M. van Laarhoven and an Innovative Scheme (Veni) grant (451-15-019) of the Netherlands Organization for Scientific Research (NWO), granted to A.I.M. van Laarhoven. H. Holle was supported by a grant from the Psoriasis Association (award number ST2/18).

This research is supported by a Leiden University Fonds (www.luf.nl) project grant (CWB 7510 / 21‐03‐2O17 / dDM ) to A.I.M. van Laarhoven and an Innovative Scheme (Veni) grant (451-15-019) of the Netherlands Organization for Scientific Research (NWO, www.nwo.nl), granted to A.I.M. van Laarhoven. H. Holle was supported by a grant from the Psoriasis Association(www.psoriasis-association.org.uk) (award number ST2/18, ). The funders had no role in study design, data collection and analysis, decision to publish or preparation of the manuscript. 

Reviewers' comments:

Reviewer's Responses to Questions

**Comments to the Author**

1. Is the manuscript technically sound, and do the data support the conclusions?

Reviewer #1: Yes

2. Has the statistical analysis been performed appropriately and rigorously? 

Reviewer #1: Yes

3. Have the authors made all data underlying the findings in their manuscript fully available?

Reviewer #1: No

4. Is the manuscript presented in an intelligible fashion and written in standard English?

Reviewer #1: Yes

5. Review Comments to the Author

Reviewer #1: Thank you for giving me the opportunity to review the paper „No preconscious attentional bias towards itch in healthy individuals“ by Becker and coworkers. The study investigated in 127 healthy persons whether a preconscious attentional bias exists to itch stimuli and whether this is modified by the presentation of mild itch stimuli, which were presented beforehand. It was shown that attention was drawn towards the neutral stimuli controversy to the hypothesis. This is interesting. However, some points need to be addressed in order to improve the paper. The main points are that it is a pity that the sample only consisted of healthy subjects without chronic itch with most of them being female. Another point is that a baseline assessment of itch is missing as far as I see and that the itch stimuli only induced a very mild itch. Here are some more specific points:

1. Data availability statement: You state that some restrictions will apply. What do you mean with this? Please specify.

2. Abstract: Please clarify in the abstract already how itch was induced (what kind of stimuli were used) and how cognitive flexibility and attentional inhibition were operationalized and measured?

3. Has the study protocol been published somewhere? Please give the trial registration number.

4. Methods:

a. How could it happen that 1 person was tested twice and how did you find out? Was data collected in a pseudonomized manner?

b. Why were persons stratified for handedness? Please explain why this was necessary to control for. Why were responses given with the index fingers of both hands?

c. How much monetary reimbursement was given to the participants? How could this have biased the results? Please consider.

d. How was randomization to the itch stimuli and control stimuli conducted? Who conducted randomization?

e. Have the mechanical stimuli been shown to induce itch before? How intense was the itch induced by this method in healthy subjects in former studies?

f. Why did you come up with new auditory control stimuli and did not use the ones used before?

g. Why was it not possible to calculate indicator of total fixation duration for this study?

h. Was there a reason to apply this order of filling in the questionnaires? Why were anxiety, depression and strtess measured before the experiment and the other variables not? I assume there was a reason. Please state. Also, I wonder why you applied all these questionnaires? Were there explicit hypotheses regarding all this variables? If yes, please state at the end of the introduction. Otherwise, please state that they were just used in an explorative manner without having specified hypotheses.

i. Page 12, line 260: why do you say mechanical vs. auditory? Does it not have to be mechancial and auditory?

5. Results

a. There were much more females in the study than males. Please conduct separate analyses for females only in an explorative manner to see whether this changes the results. This would be interesting. Also, please analyse whether woman and men differed in the questionnaire data and induced itch by mechanical and auditory stimuli.

b. These are very small differences between itch in the priming group and control group. Were these low differences also observed in former studies? Why did you decide to use only mild stimuli and not stimuli that provoked an increase of at least 2 on a NAS (0-10)? Just because a result is statistically significant, it does not mean that is really meaningful in terms of content. The effect sizes illustrate a small effect. Please consider. Also, what was the itch baseline in both groups? Was this not measured? This would be a real limitation and should be discussed.

c. Awareness: What does a mean accuracy of .49 mean?

6. Discussion

a. Were patients with any itchy conditions (even moskito bites) excluded? Were groups (priming and control group) comparable regarding baseline itch? This comparison is necessary in order to secure that the priming actually worked.

b. Regarding references 27 and 52: Were participants not blinded about the real purposes of the studies and debriefed afterwards?

c. Please, clearly state that patients with chronic itch and healthy skin controls should differ in their attention towards itch stimuli. I would assume that patients with chronic itch show a (greater) AB towards itch stimuli. What could have prevented the AB in healthy skin controls? Might they have experienced digust? How could disgust moderate AB?

7. Table S2: What was the item on itch oft he BVS? How do you interprete a negative relationship between the BVS item on pain and the AB Index for itch?

6. PLOS authors have the option to publish the peer review history of their article (what does this mean?). If published, this will include your full peer review and any attached files.

Reviewer #1: No

---

## [Author Response · Author response to Decision Letter 0]

30 Jul 2022

Thank you very much for your valuable review. We responded to your comments point by point in the response to reviewer letter that is attached.

---

## [Editor Report · Decision Letter 1]

11 Aug 2022

No preconscious attentional bias towards itch in healthy individuals

PONE-D-22-05721R1

Dear Dr. Becker,

We’re pleased to inform you that your manuscript has been judged scientifically suitable for publication and will be formally accepted for publication once it meets all outstanding technical requirement

Editor's comments: The paper is very well written and a good read overall. I think it's now ready for publication.

Kind regards,

Michael B. Steinborn, PhD

Section Editor

PLOS ONE
---

## [Editor Report · Acceptance letter]

24 Aug 2022

PONE-D-22-05721R1 

No preconscious attentional bias towards itch in healthy individuals 

Dear Dr. Becker:

I'm pleased to inform you that your manuscript has been deemed suitable for publication in PLOS ONE. Congratulations! Your manuscript is now with our production department. 

Kind regards, 

on behalf of

Dr. Michael B. Steinborn 

Section Editor

PLOS ONE